# Development of an optimized protocol for generating knockout cancer cell lines using the CRISPR/Cas9 system, with emphasis on transient transfection

Seyed Alireza Mousavi Kahaki[1], Nayereh Ebrahimzadeh[2], Hossein Fahimi[1]*, Arfa Moshiri[2¤]*

1 Faculty of Advanced Science and Technology, Department of Genetics, Tehran Medical Sciences, Islamic Azad University, Tehran, Iran, 2 Department of Mycobacteriology and Pulmonary Research, Pasteur Institute of Iran, Microbiology Research Center, Tehran, Iran

¤ Current address: Thoracic and GI Malignancies Branch, Center for Cancer Research, NCI, NIH, Bethesda, Maryland, United States of America
* h.fahimi@iaups.ac.ir (HF); arfa.moshiri@nih.gov (AM)

**Data Availability Statement:** All relevant data are within the paper and its Supporting information.

## Abstract

The clustered regularly interspaced short palindromic repeats (CRISPR) system offers cost-effectiveness, high efficiency, precision, and ease of use compared to traditional gene editing techniques. In this study, we employed findings from prestigious investigations to develop an optimized approach for generating knockout cancer cell lines using a transient transfection method. This protocol introduces a distinctive approach that follows rigorous guidelines for designing gRNA to reduce off-target effects, a major challenge in CRISPR applications. Our step-by-step instructions allow researchers, particularly those with limited laboratory equipment and funding, as well as those undertaking CRISPR projects for the first time, to generate knockout cell lines using CRISPR technology in just ten weeks. This protocol covers all needs for enhancing various yields, such as transfection efficiency, and includes leveraging robust bioinformatics tools, conducting essential assays, isolating monoclonal cells via limiting dilution, validating knockout cells, and providing comprehensive troubleshooting recommendations. Using this method, we successfully created several new generations of colorectal cancer cell lines with monoallelic and biallelic knockouts of the epithelial cell adhesion molecule (EpCAM) gene. Our method, optimized for a wide spectrum of cancer cell lines, makes CRISPR more accessible for applications in personalized and precision medicine. It expands opportunities for novel investigations into cancer mechanisms and paves the way for potential therapeutic interventions.

## Introduction

The CRISPR/Cas9 system, an adaptive immune mechanism found in bacteria and archaea, protects the host from invasive agents such as viruses. In 2012, Jennifer A. Doudna and

**Funding:** The author(s) received no specific funding for this work.

**Competing interests:** The authors have declared that no competing interests exist.

Emmanuelle Charpentier introduced the gRNA-programmed Cas9-based technique for gene editing [1], a groundbreaking advancement that earned them the 2020 Nobel Prize in Chemistry for their pioneering work in genome editing [2]. Before CRISPR, genome editing relied on tools such as ZFNs, TALENs, and meganucleases. The simplicity, cost-effectiveness, efficiency, versatility, and flexibility of CRISPR make it a superior tool for genome editing compared to these earlier methods. The CRISPR system consists of an endonuclease protein and a guide RNA (gRNA), forming a ribonucleoprotein (RNP) complex capable of independently cleaving both DNA strands. In contrast, ZFNs and TALENs use different recognition elements and FokI endonucleases, which require dimerization to cleave DNA strands [3–5]. If the active CRISPR/Cas9 complex simultaneously recognizes the protospacer adjacent motif (PAM) sequence and the complementary target sequence of the gRNA on the target DNA, it will be able to cut the target region. The wild-type Cas9 (WT-Cas9) enzyme's HNH and RUV-like cleavage domains induce double-strand breaks (DSBs) by cutting in the non-targeted (gRNA noncomplementary) and targeted (gRNA complementary) strands, respectively [5]. Advancements in CRISPR have led to modified versions, such as Cas9 nickase (nCas9), which contains specific mutations (D10A or H840A) that inactivate one cleavage domain. This variant retains DNA-binding ability but cannot cut both strands simultaneously, allowing for precise targeting without inducing DSBs. Furthermore, Cas13, also known as C2C2, functions as an endonuclease, cleaving single-stranded RNA (ssRNA) to achieve gene knockdown instead of knockout [6].

Off-target activity, characterized by unintended cleavage and alterations in non-targeted genomic regions due to sequence similarity with the target site, remains a significant challenge in CRISPR/Cas9 applications. Studies have shown that these off-target effects can be substantially reduced by employing effective strategies, including the design of highly precise gRNAs, the use of high-fidelity Cas9 enzymes, the incorporation of the nCas9 enzyme, and the selection of appropriate delivery platforms. For example, studies have demonstrated a 50–1500-fold reduction in off-target activity by using two nCas9 enzymes instead of WT-Cas9, highlighting the efficacy of these approaches in minimizing off-target effects [6,7].

This study emphasizes two strategies to minimize off-target effects: the use of stringent criteria for designing high-quality gRNAs and transient transfection. Transient transfections, such as those using plasmids, lack the ability for genomic integration that allows temporary Cas9 expression, reducing the chance for off-target effects. In contrast, stable transfections, such as lentiviral vectors, tend to integrate into the host genome in a relatively random manner, often near actively transcribed genes or within open chromatin regions, leading to continuous expression of Cas9 and gRNA. This increases the risk of off-target effects and insertional mutagenesis [8]. Professional gRNA design, employing robust bioinformatic tools, further minimizes off-target effects by ensuring high specificity and reducing potential binding to unintended genomic sites [9–13]. Previous studies have indicated that transient methods offer a distinct advantage in enhancing safety by minimizing off-target risks due to their short-term expression profile. The following sections of this study provide comprehensive details on these strategies.

## Development of the protocol

We have developed this protocol specifically for generating knockout cancer cell lines utilizing the transient transfection method. This is an updated and specialized version of the previous instructions for CRISPR/Cas9 system applications published in 2013 [14]. While the previous protocol outlined general guidelines and suggested various methods for conducting CRISPR projects, it lacked essential guidance, especially for researchers undertaking CRISPR projects

for the first time. In contrast, our comprehensive protocol provides step-by-step instructions and detailed troubleshooting solutions specifically tailored for generating knockout cell lines. This method involves the transient transfection of the pX459 vector using lipid nanoparticles. Our protocol includes guidance on leveraging robust databases, conducting the MTT assay, employing single-cell isolation technique, validating gene knockouts, and more. We are confident that researchers with limited laboratory facilities and minimal expertise can effectively utilize this protocol to achieve gene knockout in various cell lines.

## Comparison with other CRISPR genome editing methods

One crucial step in generating knockout cells is selecting the type of transfection—either stable or transient. Manipulation of living cells can potentially trigger a stressful response that impacts subsequent studies, particularly in clinical contexts. Transient transfection is an effective method that offers more rapid results and induces fewer stressful responses compared to stable transfection [8,15]. We have successfully validated an efficient gene knockout protocol utilizing the cutting-edge CRISPR/Cas9 system, demonstrating that stable transfection is not mandatory for many knockout applications, except under specific goals or circumstances. Our approach involves introducing recombinant pX459 plasmids, which carry puromycin as a selectable marker for screening post-transfection, into mammalian cell lines using Lipofectamine 3000, a safe and non-toxic transfection method. The transient nature of this approach enables swift degradation of the pX459 plasmid by cell endonucleases, identified as foreign elements in mammalian cells. However, it's essential to consider the potential integration of plasmid fragments into the mammalian cell genome at the cleavage site. While utilizing viral vectors is prevalent in genome editing studies [16,17], they exhibit limitations. First, the CRISPR system fragment expressed via viral vectors might integrate into the host genome at unknown sites, potentially causing genetic damage. Second, the integration of the CRISPR system into the host genome within lentiviral-transduced cells can result in overexpression of Cas9 and sgRNA, elevating cellular stress and the risk of off-target effects. Third, some viral vectors cause frequent immunogenic responses, leading to cellular inflammation, which can threaten cell viability. The RNP platform often entails higher costs than the plasmid platform, primarily due to the requirement for a considerable amount of purified Cas enzyme and synthesized gRNA. This poses a challenge for studies relying on RNPs. On the contrary, plasmids maintain their preference in diverse CRISPR applications due to their cost-effectiveness, ease of use, versatility across a spectrum of applications, and the inclusion of selectable markers such as green fluorescent protein (GFP) or puromycin [18,19].

## Limitations of using this protocol

Plasmids, RNPs, and RNAs are common platforms utilized in CRISPR applications. Plasmids can be independently employed in nonviral delivery methods for transient cell transfection or combined with specific plasmids within viral delivery approaches to generate complete viruses. In contrast, RNPs and RNAs are exclusively utilized for transient transfection due to their inability to integrate into the host genome. The plasmid platform includes inherent limitations that may pose challenges for specific projects. To achieve the best result, scientists must meticulously evaluate project specifications and choose the CRISPR platform that aligns best with their requirements, considering factors such as efficiency, specificity, and the potential risk of off-target effects.

First, unlike RNPs and RNA, plasmid fragments possess the capability to integrate into the host genome. Mammalian cells recognize plasmids as foreign factors and trigger the activation of nucleases against them after entry. When the non-homologous end joining (NHEJ)

mechanism performs repairs on the damaged DNA at the cleavage site within the on-target region, it can facilitate the insertion of free plasmid fragments generated by cellular nucleases into the host genome. Hence, the likelihood of plasmid-derived insertions is expected to rise [20]. For instance, in this study's context, during the sample quality assessment before undertaking Sanger sequencing, an interesting observation was made within one of the monoclonal populations. Gel electrophoresis of the polymerase chain reaction (PCR) product for the targeted region of the *EpCAM* gene revealed the presence of two bands. Notably, one band was positioned at 180 bps, while the other was situated between 500–600 bps. Subsequent analysis of the Sanger sequencing data illustrated the insertion of a 351 bps Cas9 protein-coding segment at the on-target cleavage site within one of the alleles.

Second, the CRISPR plasmid platform potentially induces more stress than the RNP platform. When DNA enters cells, it often triggers stress responses. For example, the presence of plasmid could activate cyclic GMP-AMP synthase (cGAS) in human stem cells. Kim et al. reported that the RNP platform resulted in a twofold increase in the number of living transfected embryonic stem cells compared to the CRISPR plasmid platform [20].

Third, The plasmid platform exhibits a higher potential for off-target effects than the RNP platform. Selecting a target sequence with a unique complementary position throughout the genome is one of the most crucial design criteria for gRNAs. However, achieving this uniqueness is challenging in certain instances. The similarity between sequences elsewhere in the genome and the designed gRNA correlates positively with off-target effects, particularly with the similar sequence having the fewest mismatches with the seed region (10–12 bps adjacent to the PAM) of the gRNA. The long-lasting effect of CRISPR complexes and the high concentration of CRISPR complexes in cells could increase off-target activity. Unlike RNP complexes, plasmids are more stable and can generate active CRISPR complexes after cleavage at the desired site, persisting until removal from the host via cellular DNA nuclease activity. Liang et al. reported that the ratio of off-target to on-target activity increased 28-fold when CRISPR plasmids were used instead of RNPs [20–22].

Fourth, studies suggest that the plasmid platform may demonstrate lower efficiency when compared to the RNP platform. This discrepancy arises from requiring the host transcription and translation machinery in the plasmid platform. Consequently, the formation of the active CRISPR complex might be reduced effectively, given its formation and functioning are across two different cellular locations—the cytoplasm and nucleus. Correct folding of the Cas9 protein within the host becomes pivotal for the efficacy of the plasmid platform. If folding of the Cas9 protein does not occur correctly within the host, the plasmid platform will be less effective than RNP. The RNA platform faces the challenge of rapid degradation of gRNA and mRNA due to the inherent instability of RNAs. In contrast, the presence of Cas9 protein plays a crucial role in safeguarding gRNA from degradation products, thereby ensuring the efficiency of the RNP platform. Using synthesized gRNAs with specific chemical modifications in the RNP platform will protect the gRNA from nuclease activity and limit immunogenicity and toxicity in transfected cells. Also, the long half-life of RNP compared to RNA increases the efficiency of the RNP platform. Research outcomes consistently demonstrate the superiority of editing efficiency in stem cells, primary cells, and immortalized cells when utilizing the RNPs platform over the plasmid platform [20–25].

## CRISPR/Cas9 delivery methods

Scientists can use either viral or nonviral delivery methods depending on the characteristics of their studies. Choosing an efficient strategy is critical because each method has advantages and disadvantages. Lentiviral, adenoviral, and adeno-associated virus (AAV) vectors are the most

commonly used viral vectors for gene transduction into target cells. Lentivirals are efficient vectors that exhibit high-fidelity gene transfer via intact integration and the absence of chromosomal rearrangements. Nonetheless, safety concerns arise regarding insertional mutagenesis via integration, where integration occurs in proximity to proto-oncogenes. Moreover, integrating the CRISPR coding system into the host genome heightens the risk of potential off-target activity [8]. Adenoviral vector genomes boast genetic stability and a remarkable capacity for delivering large therapeutic genes without insertional mutagenesis ability. However, concerns persist regarding the host immunogenic responses, which can potentially lead to inflammation, which remains a significant safety consideration. AAVs exhibit notable efficiency and fidelity in in vivo gene transfer. Yet, their frequent immunogenicity, along with concerns about potential insertional mutagenesis and minor chromosomal rearrangements, can pose challenges to their widespread use. Physical and chemical approaches belong to nonviral delivery methodologies [15]. Microinjection is a physical method that involves the controlled introduction of DNA, RNA, or protein into a target cell. This method is challenging due to its technical complexity and cannot be used in in vivo studies. Electroporation stands as another readily applicable physical method for in vitro applications. Nonetheless, its principal drawbacks encompass high cell mortality rates and the impossibility of performing in vivo applications. Chemical methods are mainly categorized into synthetic lipid-based, gold nanoparticle-based and polymer-based approaches. The Food and Drug Administration (FDA) has approved cationic lipid-based approaches for clinical trials since they do not contain virus particles and do not place excessive stress on the cells. Time-consuming, rigorous optimizations and different performances in various cells are the main drawbacks of this approach [26,27].

## Overview of the protocol

Fig 1 illustrates the flowchart of the steps involved in generating and validating CRISPR/Cas9--mediated knockout cancer cell lines. This protocol provides step-by-step instructions for each section and suggests alternative approaches for certain steps to enhance flexibility. We offer comprehensive troubleshooting for steps where potential issues may arise, expediting the study and eliminating the need for additional problem-solving resources. The process begins with designing an effective gRNA and culminates in verifying the knockout through the analysis of single-cell colonies using sequencing techniques. Fig 2 includes a Gantt chart indicating the timing of this process, which may vary in some parts due to specific factors such as cell characteristics and the researcher's proficiency level.

## Experimental design

**1. sgRNA design**. The success of CRISPR studies largely depends on the design and selection of the most appropriate gRNAs. Designing an appropriate gRNA in some genomic regions is problematic for several reasons. This step becomes even more crucial, particularly in clinical trials. The primary priority in selecting a gRNA should always be the absence of harmful off-target effects. For example, In cases where scientists encounter detrimental off-target activities using the wt-Cas9 enzyme, employing two gRNAs with the nCas9 enzyme emerges as one of the most effective solutions. As a result, off-target activities caused by a single nCas9 are not dangerous and can be repaired error-free and efficiently based on the opposite DNA strand [6]. In this protocol, we employ various gRNA designing tools, which utilize distinct algorithms, to design outstanding gRNAs within the target sequence. In certain CRISPR studies, like gene knockouts, the necessity arises for gRNAs to be capable of cleaving within CDS regions. However, this does not mean that the sequences submitted to the design tools will contain only the CDS, as we encounter sequences with exon-exon junctions that are not

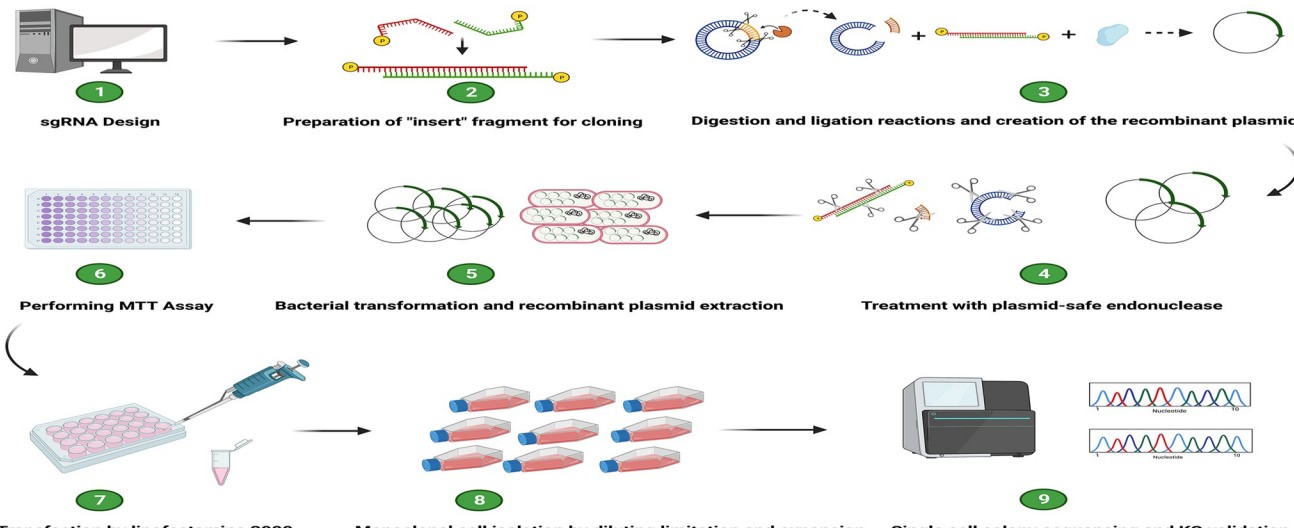

**Fig 1. Flowchart for generating CRISPR-mediated knockout cancer cell lines using transient transfection.** This flowchart illustrates the nine steps involved in generating knockout cancer cell lines. Part 1: Designing an Effective gRNA (Steps 1–7) This critical step involves adhering to several important criteria and using robust bioinformatics tools to select a gRNA with no predicted harmful off-target effects. Part 2: Preparing the Insert Fragment for Cloning (Steps 8–13) This part involves annealing two oligonucleotides (the designed sequence and its complement) to prepare the insert fragment for the ligation reaction. Part 3: Digestion and Ligation Reactions for Producing Recombinant Plasmids (Steps 14–23) We use the BbSI restriction enzyme to remove the spacer segment and create a recombinant vector through a ligation reaction between the prepared insert fragment and a linearized vector using T4 DNA ligase. Part 4: Treatment with Plasmid-Safe Endonuclease (Steps 24–25) This optional part eliminates any remaining linearized vectors that did not participate in the ligation reaction, thereby increasing transformation efficiency. Part 5: Bacterial Transformation and Recombinant Plasmid Extraction (Steps 26–37) This step increases the amount of recombinant DNA. It is critical to obtain high-quality recombinant plasmids that are free of contamination, such as LPS, because it can have a significant impact on transfection efficiency. Part 6: Performing the MTT Assay to Determine Optimal Puromycin Concentration (Steps 38–44) Selecting the correct puromycin concentration is vital. An abnormal concentration can result in the death of transfected cells or the survival of untransfected cells. Part 7: Transfection of Recombinant Plasmids into Target Cells (Steps 45–57) We use cationic lipid nanoparticles for transient transfection. These nanoparticles encapsulate the recombinant plasmids and deliver them to the target cells. Part 8: Monoclonal Cell Population Isolation by Limiting Dilution (Steps 58–69) This lengthy process requires precise execution. Each well must receive either one or zero cells; any other outcome will result in a polyclonal cell population, invalidating the results. Part 9: Single-Cell Colony Sequencing Analysis and Knockout Validation (Steps 70–74) In the final part, we compare the sequencing traces between monoclonal cell lines and the wild-type population. Trace alignment should show divergence in the target region and frameshift deletion for knockout validation in both alleles. Created with BioRender.com.

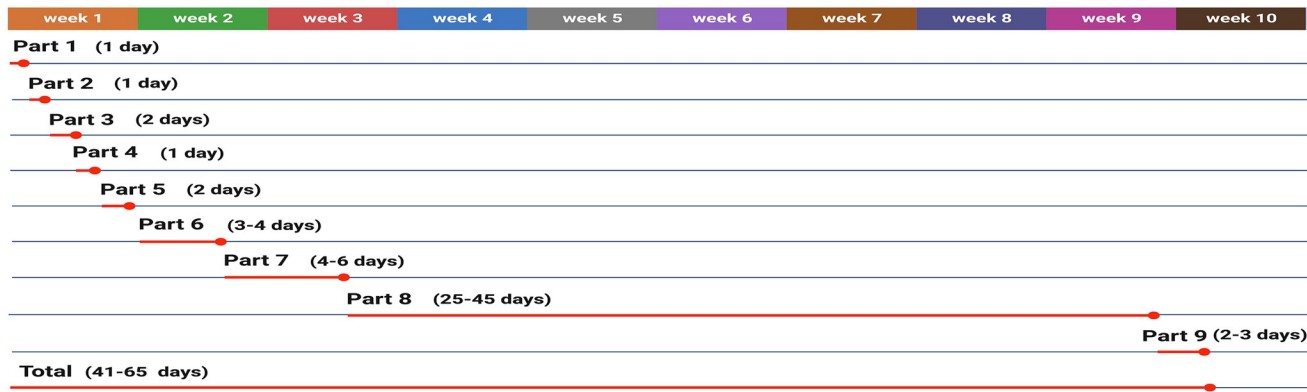

**Fig 2. The Gantt chart for the timing of this protocol.** The Gantt chart illustrates the projected timeline for each part and the anticipated duration required to execute this protocol, focusing on transient transfection of the pX459 vector using lipid nanoparticles to induce a genetic knockout in cancer cell lines via CRISPR/Cas9. It's important to note that the provided timeline is an estimation and may vary due to factors such as the specific cell lines utilized, the proficiency of the researchers, and any unforeseen challenges encountered during the procedure. Created with BioRender.com.

present in genomic DNA. Consider the following parameters to select an appropriate gRNA for gene knockout.

1. The gRNA should recognize the most common exons of the different transcripts of a gene.

2. The 20-nt target sequence should be placed in the earliest exons of the coding regions.

3. The optimal GC content for the target sequence is between 40–60% because a high GC content can lead to robust secondary structures in gRNAs, reducing the efficiency of active ribonucleoprotein complex formation.

4. The number and position of mismatches within a gRNA indicate different off-target activities. Mismatches outside the seed region increase the risk of off-target cleavage. Therefore, gRNAs with this feature should be excluded from the list of candidates.

5. The gRNA must have high Insertion–deletion (InDel) efficiency when evaluated with multiple algorithms.

6. The 3'-ends of the target sequence of the gRNA (especially the four nucleotides upstream of the PAM sequence) should not contain the 'TT-' and 'GCC-' motifs. The 'TT-' motif decreases the amount of active ribonucleoprotein complex due to the lower expression of gRNAs. The 'GCC-' motif increases the possibility of nonspecific binding to potential off-targets, and the GC-rich sequence may obstruct proper PAM interaction [28].

7. We strongly recommend using gRNA with at least one "G" as the first nucleotide at the 5'-end instead of manually appending "G" to the selected gRNA to increase the expression efficiency of the U6 promoter.

8. When off-target effects are predicted in critical regions, e.g., at the regulatory sites for gene expression of oncogenes and in the coding regions of tumor suppressor genes, it is necessary to design primers for these regions for sequencing to ensure that nothing has been altered [9–13].

**2. Preparation of the 'insert' fragment for cloning**. The process of synthesizing the insert fragment necessitates several modifications and manipulations of the 20-nt target sequence. The BbsI restriction enzyme performs two cleavages in the pX459. In one cleavage, the upper strand is cleaved 4-nt before at the end of the U6-promoter coding region, and the lower strand is cleaved precisely at the end of the U6-promoter coding region, resulting in a 4-nt overhang in the pX459. In another cleavage, the upper strand is cleaved precisely before the gRNA-scaffold sequence, and the lower strand is cleaved 4-nt downstream of the upper strand cleavage point, resulting in another 4-nt overhang in the pX459. As a result, both of the four specific nucleotides should be added manually to the insert fragment. To complete the ligation reaction, phosphorylation of the 5'-ends of strands is critical. During the digestion reaction, the phosphorylation status of backbone 5'-ends is retained. For the ligation reaction, phosphorylation of both insert fragments 5'-ends is crucial. There are two solutions for this problem: first, employing chemically modified oligonucleotides (using synthetically phosphorylated oligonucleotides), which run the process without any extra steps, although at a higher cost than unmodified oligonucleotides. Second, treatment with T4 poly nucleotide kinase (T4PNK). Inactivation of T4PNK must be performed by heating to prevent inhibition of subsequent enzymatic reactions. Fig 3 illustrates the annealed insert fragment, its location within the pX459 plasmid construction, and the spacer fragment that will be released after the digestion reaction.

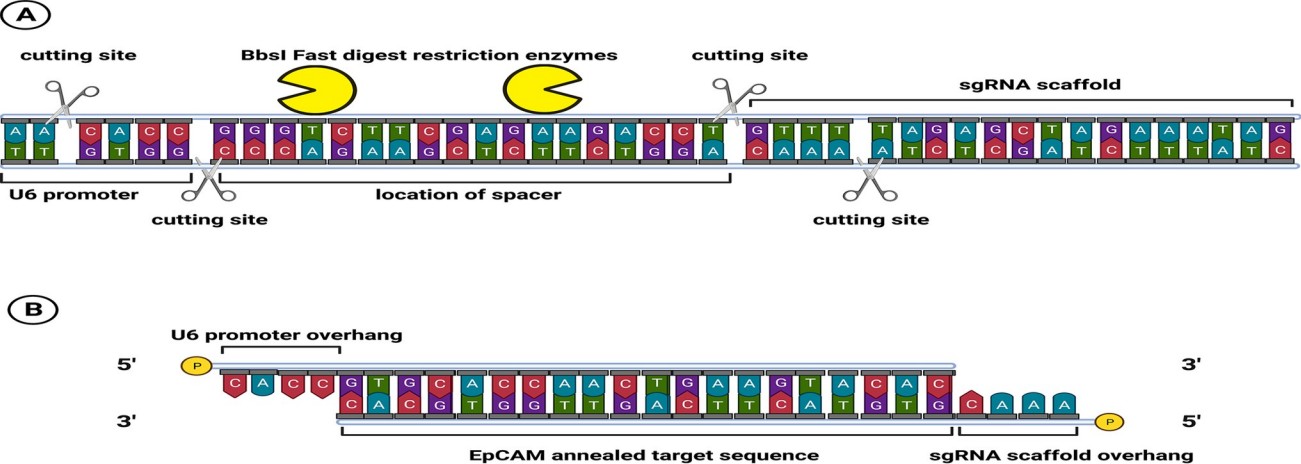

**Fig 3. The schematic picture for the recombinant *EpCAM*-pX459 plasmid construction.** A) The pX459 vector contains two recognition sites of the BbsI enzyme, indicated by yellow symbols, located at the spacer site. The scissor symbols represent the specific cutting sites where the BbsI enzyme cleaves the DNA. This enzymatic action releases a 22-bp fragment with two 4-nt sticky ends. B) The top and bottom strands are annealed at a precise temperature. The annealed strand comprises 20 nucleotides from the target sequence specifically designed for the *EpCAM* gene. The two 4-nt fragments, previously excised by the restriction enzyme, are manually reattached to the 5'-end of their respective strands. Notably, during oligonucleotide synthesis, the 5'-ends of both strands were phosphorylated to facilitate subsequent steps in the cloning process. Created with BioRender.com.

- **POINT:** Biotechnology companies develop standard buffers in which most enzymes are active. We strongly recommend using enzymes from the same manufacturer for digestion, ligation, and phosphorylation. Since the phosphorylation reaction uses gamma-phosphate of ATP and the digestion buffers do not contain ATP, ATP must be added to the reaction components for T4PNK treatment after the digestion reaction is complete. At the same time, the ligation buffer includes the required amount of ATP, so no additional ATP is needed. In other words, digestion, phosphorylation, and ligation reactions can be performed with a ligation buffer.

   **3. Digestion and ligation reactions and creation of the recombinant plasmid**. The cloning of the insert fragment into the pX459 vector can be achieved through two methodologies. The first approach involves a sequential two-step process encompassing digestion followed by ligation, requiring gel extraction of the digested product. In contrast, the second strategy streamlines this process by combining digestion and ligation into a single step involving two consecutive reactions, differing from the first method in key aspects. In the second approach, if alkaline phosphatase treatment is not performed before ligation, competition arises between two DNA fragments, the 22-bps spacer, and the insert fragment, in the ligation reaction. The thermosensitive alkaline phosphatase enzyme removes phosphate from the entire 5'-ends. As a result, the 22-bps spacer fragment won't be able to participate in the ligation reaction, while the 5'-ends of the insert fragment are phosphorylated and cloned into the vector. Despite an incomplete ligation since the 5'-ends of the vector are unphosphorylated, post-transformation bacterial repair mechanisms repair the nicks, resolving the problem.

   **4. Treatment with plasmid-safe endonuclease (optional)**. The plasmid-safe endonuclease enzyme efficiently digests linear DNA but does not affect nicked, closed double-stranded DNA or supercoiled DNA structures like plasmids or fosmids. While not mandatory, this enzymatic step could enhance the efficiency of the transformation process by eliminating any residual linearized plasmids resulting from the ligation reaction. When linear DNA fragments

enter a prokaryotic cell, it triggers the activation of systems responsible for their degradation. Interestingly, experimental results indicate that the presence or absence of linear DNA plasmids does not fundamentally influence the outcome of the transformation process.

**5. Bacterial transformation of recombinant plasmid extraction**. The primary goal of this section is to amplify the copy number of recombinant *EpCAM*-pX459 plasmids through their transfer into competent bacteria, which can be performed using conventional bacterial transformation methods, specifically heat shock or electroporation. Our study focuses on the optimized heat-shock technique to efficiently transform pX459 plasmids into *DH5-alpha-competent Escherichia coli (E. coli) cells*. The volume of competent bacteria required is contingent upon the volume of the ligation reaction or the output from plasmid-safe endonuclease treatment, typically necessitating approximately 10-times the volume of the ligation reaction (or approximately 7-times the volume of the plasmid-safe endonuclease treatment). This protocol uses a specific PCR validation procedure. The forward primer is positioned on the backbone within the U6 promoter region preceding the 20-nt target sequence, while the reverse primer targets both the backbone and the target sequence. Successful ligation manifests as a distinct 270-bps band within the PCR product, serving as a definitive indicator of the desired recombination event.

**6. Performing MTT assay**. Mitochondrial reductases convert MTT to formazan, a purple insoluble precipitate, in living cells. Conducting the MTT assay on the desired cell line is crucial for establishing the optimal concentration of puromycin antibiotic for screening purposes. In the pX459 structure, the T2A translational linker facilitates the connection between the nuclear localization signal (NLS) tag at the C-terminal domain of Cas9 and the N-terminal end of the puromycin coding region. In the absence of the puromycin resistance gene, untransfected cells may exhibit resistance to the antibiotic at low concentrations. For screening adherent cell lines, puromycin concentrations are typically higher than for suspension cell lines. This protocol uses transient transfection to knock out the target gene. Therefore, the calculated concentration and duration of puromycin treatment for the same cell line may differ between this method and the stable transfection technique. High cell confluency indeed presents challenges in assessing puromycin uptake during screening. When cells densely interconnect due to high confluency, the penetration and diffusion of puromycin may be impeded, affecting its efficacy in eliminating non-resistant cells. This condition complicates the determination of the optimal antibiotic concentration and treatment duration. Researchers may face difficulties in accurately evaluating the impact of different puromycin concentrations due to uneven distribution and reduced accessibility of the antibiotic to all cells within a high-density culture. Adjusting seeding densities or optimizing culture conditions may help mitigate these challenges and ensure more reliable assessments of puromycin's effects on the cell population. To perform the MTT assay, a specific number of cells should be seeded so that the number of cells available in the MTT assay at the time of screening after transfection is approximately equal to the number of cells available in the negative control. Puromycin concentrations are classified into one of the following categories:

**The high dose range of puromycin:** Within less than 24 hours of puromycin treatment, a concentration of puromycin can kill all living cells with metabolic activity. Consequently, this concentration is often excessive, surpassing the tolerance threshold of cells containing the puromycin-resistant gene, leading to cell death and reduced transfection yield.

**The optimal dose of puromycin:** During the 48- or 72-hour puromycin treatment, a concentration of puromycin is sufficient to kill all untransfected cells with metabolic activity. However, on the other hand, cells that received a resistance gene during transfection can tolerate this concentration.

**The low dose range of puromycin:** After 72 hours of puromycin treatment, a concentration of puromycin can keep all or some untransfected cells alive and allow them to continue their metabolism. Typically, this concentration is low enough that normal cells can tolerate it and live alongside transfected cells; as a result, researchers will obtain false-positive results.

- **POINT:** The manufacturer of Lipofectamine recommends a minimum confluence of cells seeded for the transfection process of approximately 70% at the time of transfection. Increasing cell confluence above 70% may decrease the amount of puromycin taken up, increasing the time required for treatment with the antibiotic puromycin during screening. In developing this protocol, we have attempted to match the parameters of these two parts as closely as possible.

**7. Transfection using lipofectamine**. Transfection using lipofectamine is a pivotal step in this protocol. Cationic lipid reagents have become standard for transferring DNA and RNA into cells, overcoming limitations in transfecting challenging cells like primary cells, stem cells, or certain cancer cell lines. However, several factors significantly affect transfection efficiency:

- Using low passage number cell lines

- Avoiding penicillin-streptomycin antibiotics during transfection

- Ensuring the absence of lipopolysaccharide (LPS) contamination in the extracted recombinant plasmid

- Attaining optimal cell confluence percentage

- Increasing serum amount in the cell culture medium for transfection

- Employing a serum-reduced medium for lipofectamine-plasmid complex preparation

- Refreshing culture medium in wells before transfection

- Avoiding immediate transfection after cell thawing

These factors may require optimization depending on the cell lines used by researchers. Expanding transfected cells before proceeding with monoclonal cell isolation is crucial for several reasons: Firstly, the expansion allows DNA extraction for subsequent Sanger sequencing analysis, verifying the CRISPR system's correct function in the target region. Secondly, maintaining a population backup becomes essential to mitigate potential future issues. Expanding cells ensures continued access to the initial population as a valuable resource. Thirdly, achieving accurate and error-free dilutions for isolating single cells necessitates a minimum cell number, ensured through cell expansion. This step guarantees the precise and reliable isolation of individual cells.

**8. Monoclonal cell isolation by limiting dilution and expansion**. In this method, 96-well plates are utilized for isolating single cells. Each well should contain 100 μl of culture medium, and determining the appropriate cell concentration for the final dilution is crucial. Our objective is to distribute one cell per well or, more precisely, one cell per 100 μl of culture medium. However, due to the uneven distribution of cells in the medium, selecting one cell per well may result in wells containing multiple cells, which is undesirable for our purposes. We recommend a concentration range of 0.4 to 0.6 cells per 100 μl to address this issue. This range balances the trade-off between accuracy and the occurrence of multiple cells in a well. At a concentration of 0.5 cells per 100 μl, the probability of finding a cell in a well is approximately 50%, ideally resulting in about half of the wells containing a single cell. It is important to note

that the uneven distribution of cells may lead to some wells having more than one cell or no cells at all. Once the cell confluence reaches an acceptable percentage, the expanded monoclonal cell population in each well must be harvested meticulously to prevent cross-contamination with cells from other wells. Subsequently, the cells are grown in T-25 flasks for storage and further analysis.

**9. Single-cell colony sequencing and knockout validation**. After DNA extraction from expanded monoclonal cell populations, it is necessary to amplify the CRISPR target region using PCR. The gRNA design software suggests PCR primers for each target sequence. Upon assessing the quality of PCR products by gel electrophoresis, Sangar sequencing should be performed with one of the primers. Except for deletions of multiples of three in the nucleotide sequence, all deletions can result in frameshift mutations. If the number of deleted nucleotides is multiples of three in a way that interferes with more than one codon, this can lead to the formation of immature stop codons.

## Materials and methods

### Materials

1. **Vector construction and molecular technique**

1.1) pSpCas9(BB)-2A-Puro or pX459 (Addgene, plasmid ID: 48139)

1.2) PCR primers and oligos for sgRNA construction are listed in Table 1. Primers were ordered as 4-nmol ultramers (Bioneer Company)

1.3) Fast Digest BpiI (BbsI) (Thermo Scientific™, Catalog No. FD1014)

1.4) T4 DNA Ligase (5 U/µl) (Thermo Scientific™, Catalog No. EL0014)

**Table 1. List of PCR primers and oligonucleotides.**

| Step (s) | Name | Sequence (5′–3′) | Length | Purpose |
|---|---|---|---|---|
| 6 | 20-nt target sequence of sgRNA | GTGCACCAACTGAAGTACAC | 20 | Synthesis of sgRNA-top-EpCAM and sgRNA-bottom-EpCAM oligos. |
| 7 | sgRNA-top-general | CACCNNNNNNNNNNNNNNNNNNNN | 24 | Oligo for sgRNA vector construction with 20-nt sgRNA sequence for cloning of sgRNA into pX459. |
| 7 | sgRNA-bottom- general | AAACNNNNNNNNNNNNNNNNNNNN | 24 | Oligo for sgRNA vector construction with 20-nt reverse complementary sequence for cloning of sgRNA into pX459 |
| 7 8 9 | sgRNA-top-EpCAM | CACCGTGCACCAACTGAAGTACAC | 24 | Preparation of the insert fragment for cloning into pX459 |
| 7 8 9 35 | sgRNA-bottom-EpCAM | AAACGTGTACTTCAGTTGGTGCAC | 24 | 1) Preparation of the insert fragment for cloning into pX459 2) To amplify the 273bp U6 promoter-sgRNA cassette of pX459. |
| 35 | hU6 forward primer | GAGGGCCTATTTCCCATGATTCC | 23 | To amplify the 273bp U6 promoter-sgRNA cassette of pX459 |
| 54 71 | EpCAM-F. P | AAACTACAAGCTGGCCGTAAAC | 22 | To perform PCR for amplification of the on-target region |
| 54 71 | EpCAM-R. P | AAACTCTTTCCAACTCAAGGCA | 22 | To perform PCR for amplification of the on-target region |
| 56 73 | Sequencing Primer | AAACTCTTTCCAACTCAAGGCA | 22 | To perform the sanger sequencing of the on-target region |

1.5) FastAP Thermosensitive Alkaline Phosphatase (1 U/μl) (Thermo Scientific™, Catalog No. EF0654)

1.6) Hybrid-Q™ Plasmid Rapidprep (GeneAll, Catalog No. 100–150)

1.7) Expin™ Gel SV (GeneAll, Catalog No. 102–150)

1.8) Pfu DNA Polymerase Mastermix [2X] (G-Biosciences, Catalog No.786-817)

1.9) Nuclease-free water (Thermo Fisher Scientific, Catalog No. AM9930)

1.10) 1 kb DNA ladder (Yektatajhiz, Catalog No. YT8507)

1.11) 100 bp DNA Marker (Parstous, Catalog No. B111401)

1.12) Ampicillin sodium salt (Sigma-Aldrich, Catalog No. A9518-5G)

1.13) Low EEO Agarose (Yektatajhiz, Catalog No. YT9059)

1.14) DNA Green Viewer (Parstous, Catalog No. B111151)

1.15) Tris/Acetate/EDTA (TAE) Buffer, 10X Solution, Molecular Biology Grade, Ultrapure (Thermo Scientific™, Catalog No. J75904.K8)

1.16) LB Broth with agar (Lennox) (Sigma-Aldrich, Catalog No. L2897-250G)

1.17) LB Broth (Lennox) (Sigma-Aldrich, Catalog No. L3022-250G)

1.18) 5x DNA Loading Buffer Blue (Ampliqon, Catalog No. A608204)

2. **Mammalian cell culture and cellular assay and technique**

2.1) DMEM High Glucose (4.5 g/l) (Cegrogen Biotech, Catalog No. E0500-190)

2.2) Opti-MEM™ I Reduced Serum Medium (Gibco™, Catalog No. 31985062)

2.3) Fetal bovine serum (FBS), qualified, Brazil (need to heat-inactive by user) (Gibco™, Catalog No. 10270106)

2.4) Penicillin-Streptomycin (5,000 U/ml) (Gibco™, Catalog No.15070063)

2.5) Trypsin EDTA (0.25%) in HBSS with phenol red (Cegrogen Biotech, Catalog No. N0100-751)

2.6) Lipofectamine™ 3000 Transfection Reagent (Invitrogen™, Catalog No. L3000001)

2.7) Puromycin dihydrochloride from Streptomyces alboniger (Sigma-Aldrich, Catalog No. P8833-10MG)

2.8) Dulbecco's phosphate-buffered saline (DPBS), no calcium, no magnesium (Gibco™, Catalog No. 14190250)

2.9) Thiazolyl Blue Tetrazolium Bromide (MTT) (Sigma-Aldrich, Catalog No. M5655-100MG)

2.10) Dimethyl sulfoxide (DMSO), ACS reagent (Sigma-Aldrich, Catalog No. 1029521011)

2.11) Exgene™ Cell SV (GeneAll, Catalog No. 106–101)

3. **Biological materials**

3.1) HCT-116 cell line (ATCC, CCL-247)

3.2) *DH5α* chemically competent *E. coli* (Bonbiotech, Catalog No. BN-0011.7.1)

## 4. Equipment

4.1) Petri Dish, 90 mm × 20 mm (SPL life sciences, Catalog No. 10101)

4.2) 25 cm$^2$ cell culture flask, filtered cap (SPL life sciences, Catalog No. 70325)

4.3) 75 cm$^2$ cell culture flask, filtered cap (SPL life sciences, Catalog No.70375)

4.4) 96-wells plate, flat bottom and Sterile (SPL life sciences, Catalog No. 30096)

4.5) 24-wells plate, flat bottom and Sterile (SPL life sciences, Catalog No. 30024)

4.6) 40 μm cell strainer mesh (SPL life sciences, Catalog No. 93040)

4.7) 15 ml conical Tube (SPL life sciences, Catalog No. 50015)

4.8) 50 ml conical Tube (SPL life sciences, Catalog No. 50050)

4.9) 1 ml Serological Pipette (SPL life sciences, Catalog No. 91001)

4.10) 5 ml Serological Pipette (SPL life sciences, Catalog No. 91005)

4.11) 10 ml Serological Pipette (SPL life Sciences, Catalog No. 91010)

4.12) One-channel Reservoir, Sterile (SPL life sciences, Catalog No. 22001)

4.13) 1.5 ml microcentrifuge tube, Sterile (SPL life sciences, Catalog No.60115)

4.14) 0.2 ml PCR Tube (SPL life sciences, Catalog No. 60001)

4.15) Gel extractor (SPL life sciences, Catalog No. 410511)

4.16) 1.8 ml cryovial (SPL life sciences, Catalog No. 43112)

4.17) Disposable spreader (SPL life sciences, Catalog No. 90050)

4.18) Counting chamber (Neubauer-improved) (Marienfeld, Catalog No. 0640011)

4.19) 0.22 μm, Millex-GP Syringe Filter Unit (Millipore, Catalog No. SLGP033RS)

4.20) BioTek's Epoch Microplate Spectrophotometer (Agilent Technologies)

4.21) Mastercycler® nexus GSX1 (Eppendorf, Catalog No. 6345000044)

4.22) Microcentrifuge 5430 (Eppendorf, Catalog No. EP5427000216)

4.23) PowerPac Universal Power Supply (Bio-Rad, Catalog No 1645070)

4.24) Sub-Cell® GT Agarose Gel Electrophoresis Systems (Bio-Rad, Catalog No 170–4401)

4.25) DigiDoc-It® Imaging System (Analytik Jena, Catalog No 97-0243-01)

## 5. Formulation of solutions

5.1) Heat inactivation of FBS: Inactivate the FBS by heating it for 30 minutes in a water bath at 56˚C

5.2) Ampicillin solution preparation: Prepare a stock solution (50 mg/ml, pH 7.4). The final ampicillin concentration in LB broth should be 100 μg/ml.

5.3) MTT stock solution preparation: Prepare a 5 mg/ml concentration of MTT solution by dissolving 100 mg of MTT powder in up to 20 ml of sterile DPBS (1X, pH 7.4) and mixing thoroughly to achieve the desired concentration for the MTT assay. Filter the solution

through a 0.22 μm filter. Due to its light sensitivity, aliquot this solution into 2 ml sterile, dark cryotubes and store them at 4˚C.

- **POINT:** A small quantity of MTT may have remained as insoluble particles in the primary solution. These particles are removed after filtration.

5.4) Puromycin stock solution preparation: Prepare a 10 mg/ml concentration of puromycin by dissolving 10 mg of puromycin powder in up to 1 ml of sterile ddH$_2$O (pH 7.4) and label it as "primary stock solution".

Since a concentration of 10 mg/ml is too high for use, dilute the primary stock solution by one-tenth. Dilute 100 μl of the "primary stock solution" by adding 900 μl of sterile ddH$_2$O and label it as "W$_A$ (1 mg/ml) ".

Then, dilute 100 μl of the "W$_A$" by adding 900 μl of sterile ddH$_2$O and label it as "W$_B$ (0.1 mg/ml) ". Cover all dilutions using foil and store them at -20˚C.

## Methods

The protocol described in this peer-reviewed article is published on protocols.io (http://dx.doi.org/10.17504/protocols.io.5jyl82137l2w/v1) and is included for printing purposes as S1 File.

## Results

Our approach presents an innovative and pioneering method that uses the state-of-the-art CRISPR/Cas9 system to accelerate the generation of knockout cancer cell lines. We thoroughly validated the effectiveness of our method by specifically targeting the *EpCAM* gene within the HCT-116 cell line. The *EpCAM* gene encodes a transmembrane glycoprotein recognized as a cluster of differentiation 326 (CD326). Our strategic utilization of CRISPR technology focused on targeting exon 2, which codes for an EGF-like motif in its extracellular domain. We aimed to generate HCT-116 cell lines containing biallelic knockouts of the *EpCAM* gene, which has a significant expression in colorectal cancer (CRC) and established oncogenic characteristics [29]. Our investigation led to the generation of novel genotypes of the HCT-116 cell line.

In the Z14 monoclonal cell population, showcased in Table 2 and Fig 4, the NHEJ repair mechanism causes the removal of 2-nt from the cleavage site (cysteine codon) from both alleles of the *EpCAM* gene. The residual cysteine nucleotide joined to the next threonine codon (T), creating a novel codon that encodes for tyrosine (Y). This deletion resulted in the introduction of 28 premature termination codons (PTCs) within the sequence. As a result, both alleles of

**Table 2. Comparison between Sequence of amino acids for CD-326 of Z14-KO and wild-type populations.**

| Cell population | Sequence of amino acids for CD-326 |
|---|---|
| Wild-type | MAPPQVLAFGLLLAAATATFAAAQEECVCENYKLAVNCFVNNNRQCQCTSVGAQNTVICSK LAAKCLVMKAEMNGSKLGRRAKPEGALQNNDGLYDPDCDESGLFKAKQCNGTSMCWCVN TAGVRRTDKDTEITCSERVRTYWIIIELKHHKAREKPYDSKSLRTALQKEITTRYQLDPKFITSIL YENNVITIDLVQNSSQKTQNDVDIADVAYYFEKDVKGESLFHSKKMDLTVNGEQLDLDPGQ TLIYYVDEKAPEFSMQGLKAGVIAVIVVVVIAVVAGIVVLVISRKKRMAKYEKAEIKEMGEM HRELNA* |
| Z14-KO | MAPPQVLAFGLLLAAATATFAAAQEECVCENYKLAVNCFVNNNRQCQYFSWCTKYCHLLK AGCQMFGDEGRNEWLKTWEKSKT*RGPPEQ*WAL*S*LR*ERAL*GQAVQRHLHVLVCEHC WGQKNRQGH*NNLL*ASENLLDHH*TKTQSKRKTL***KFADCTSEGDHNALSTGSKIYHEY FV*E*CYHY*SGSKFFSKNSE*CGHS*CGLLF*KRC*R*ILVSF*ENGPDSKWGTTGSGSWSNF NLLC**KST*ILNAGSKSWCYCCYCGCGDSSCCWNCCAGYFQKEENGKV*EG*DKGDG*DA *GTQCI |

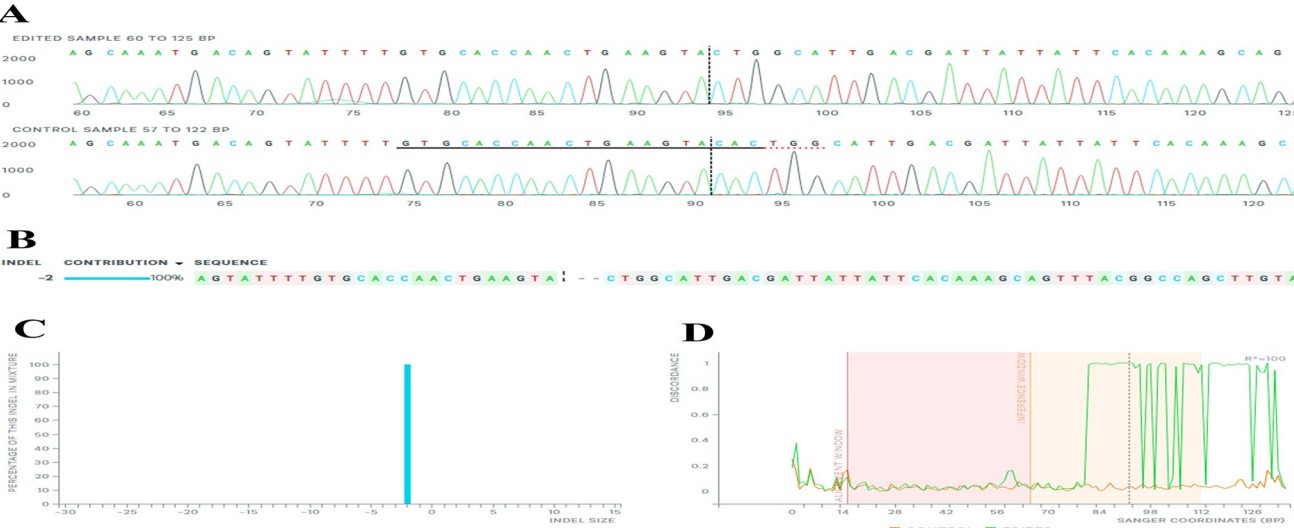

**Fig 4. The comparison between sequencing results of Z14-KO and wild-type populations.** A) Sanger sequencing traces of the Z14-KO (edited sample) and wild-type (control sample) regions around the 20-nt target sequence are shown. The area below is marked with black and red dashed lines representing the 20-nt target and the PAM sequence, respectively. The vertical black dotted line represents the cleavage site. The PAM sequence (TGG) has been intact despite deleting nucleotides downstream of the cleavage site. B) The 2-bps deletion is observed in the Z14 monoclonal population, causing a frameshift mutation in both alleles of the *EpCAM* gene. This mutation signifies that this population is a bi-allelic knockout (KO) clone, as confirmed by Sanger sequencing of the genomic DNA region around the Z14 monoclonal population's target site (edited sample), comparing it to the reference wild-type sequence (control sample). C) The InDel bar graph depicts the percentage distribution of deletions within the monoclonal Z14 genome population, categorized by InDel size. D) The alignment window refers to the region of the sequence where the alignment is performed between different sequences, and the inference window is the segment within the alignment window where the inference or assessment of discrepancies is made. The discordance plot illustrates the level of alignment per base between the wild type (control) and the monoclonal Z14 population (edited) for each nucleotide position. CRISPR editing function causes discrepancies to appear around the cleavage site, and the alignment lines remain far apart after the cleavage site, indicating a high degree of sequence discordance. ***Synthego Performance Analysis, ICE Analysis. 2019. v3.0. Synthego; [2023]***.

the *EpCAM* gene within the Z14 population encoded a truncated CD-326 protein. Fig 5 presents a comparison between Z14-KO and wild-type populations at different time points during cell culturing.

In the X2 monoclonal cell population depicted in Table 3 and Fig 6, the NHEJ repair mechanism resulted in the removal of 2-nt from the cysteine codon (C), 3-nt from threonine codon (T), and 1-nt from the serine codon (S). The remaining nucleotide of cysteine codon nucleotide merged with 2-nt of serine codon, leading to a PTC. Consequently, both alleles of the *EpCAM* gene in the X2 monoclonal cell population encoded a truncated CD326 protein.

In the X6 monoclonal cell population illustrated in Table 4 and Fig 7, the NHEJ repair mechanism causes the removal of cysteine (C) and the adjacent glutamine (Q) codons in one of the alleles of the *EpCAM* gene. As a result, one allele of the CD326 encodes proteins lacking cysteine and glutamine at the specified location. In another allele of the *EpCAM* gene within the X6 monoclonal cell population, a fragment of the Cas9 coding region, derived from the transfected plasmid as displayed in Tables 5 and 6 and Fig 7, during NHEJ repair inserted into the host genome at the cleavage site (the cysteine codon). The insertion within the middle of the cysteine codon resulted in a novel sequence that is distinct from the fragment's original coding sequence, generating a PTC. Consequently, the second allele of the *EpCAM* gene in the X6 monoclonal cell population encodes a truncated CD326 protein, identifying this cell population as a monoallelic KO clone.

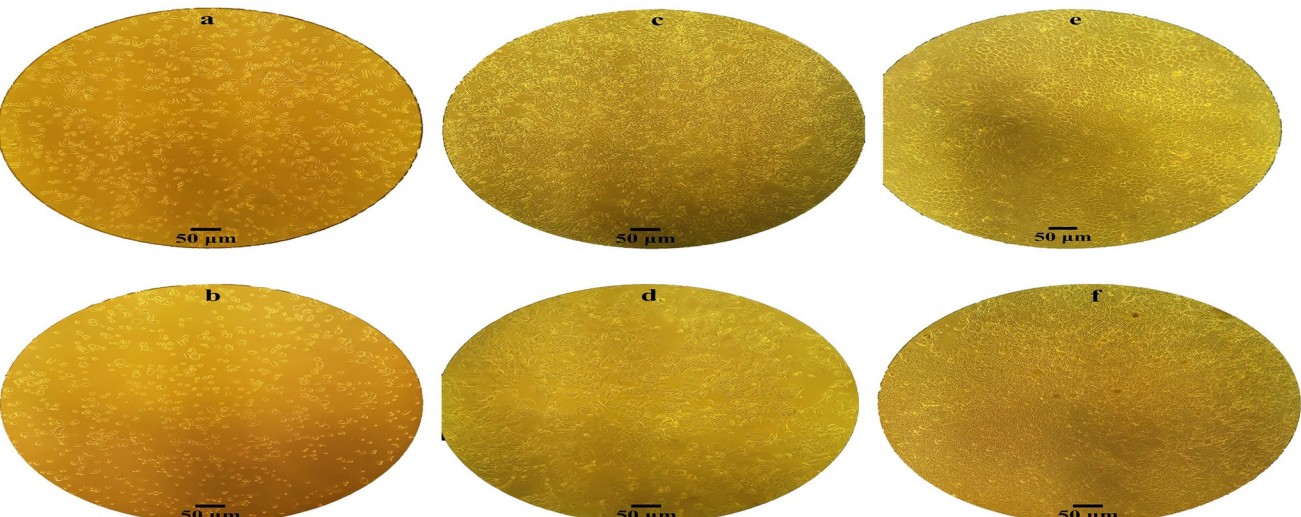

**Fig 5. Comparison of Z14-KO and wild-type populations at different time points of culturing.** After validating the bi-allelic knockout, we cultured Z14-KO and wild-type populations simultaneously to investigate differences in their rates of reaching specific confluency and mortality. Notably, the knockout population was unable to grow beyond 50–60% confluence, resulting in a dramatic increase in cell mortality at this level. Increasing the FBS percentage to 20% allowed the knockout cells to reach near full confluence for a longer period than the wild-type cells. a) The majority of cells in the wild-type population exhibited elongated morphology 24 hours after subculturing. b) Conversely, the Z14-KO population displayed a significant proportion of round cells and a minority of elongated cells 24 hours after subculturing. c) At 72 hours after subculturing, the wild-type population reached 80–90% confluence, exhibiting normal morphology. d) At 72 hours after subculturing, the Z14-KO population displayed 50–60% confluence and distinct morphological features. e) The wild-type population achieved full confluence four days after subculturing, displaying normal morphology. f) However, the Z14-KO population, observed nine days after subculturing at near full confluence, experienced a high rate of cell death before reaching this confluence stage. All observations were conducted using a 10X objective lens.

## Discussion

We indicated that using plasmid vectors, similar to some viral vectors, might pose the possibility of insertion into the host genome; however, the processes are fundamentally different. In this study, we demonstrated that a part of the Cas9 coding region was inserted at a predictable site—the Cas9 cleavage site—in the target region. On the other hand, viral vectors that integrate into the host genome do so unpredictably, which has the potential to disrupt tumor suppressor gene expression or lead to the overexpression of proto-oncogenes [8,15]. The limitations of using microinjection and electroporation for in vivo studies, along with other disadvantages mentioned earlier in this paper, make lipofectamine a favorable choice for

**Table 3. Comparison between sequence of amino acids for CD-326 of X2-KO and wild-type populations.**

| Cell population | Sequence of amino acids for CD-326 |
|---|---|
| Wild-type | MAPPQVLAFGLLLAAATATFAAAQEECVCENYKLAVNCFVNNNRQCQCTSVGAQNTVICSK LAAKCLVMKAEMNGSKLGRRAKPEGALQNNDGLYDPDCDESGLFKAKQCNGTSMCWCVN TAGVRRTDKDTEITCSERVRTYWIIIELKHHAREKPYDSKSLRTALQKEITTRYQLDPKFITSIL YENNVITIDLVQNSSQKTQNDVDIADVAYYFEKDVKGESLFHSKKMDLTVNGEQLDLDPGQ TLIYYVDEKAPEFSMQGLKAGVIAVIVVVVIAVVAGIVVLVISRKKRMAKYEKAEIKEMGEM HRELNA* |
| X2-KO | MAPPQVLAFGLLLAAATATFAAAQEECVCENYKLAVNCFVNNNRQCQ*VGAQNTVICSK LAAKCLVMKAEMNGSKLGRRAKPEGALQNNDGLYDPDCDESGLFKAKQCNGTSMCWCVN TAGVRRTDKDTEITCSERVRTYWIIIELKHHAREKPYDSKSLRTALQKEITTRYQLDPKFITSIL YENNVITIDLVQNSSQKTQNDVDIADVAYYFEKDVKGESLFHSKKMDLTVNGEQLDLDPGQ TLIYYVDEKAPEFSMQGLKAGVIAVIVVVVIAVVAGIVVLVISRKKRMAKYEKAEIKEMGEM HRELNA* |

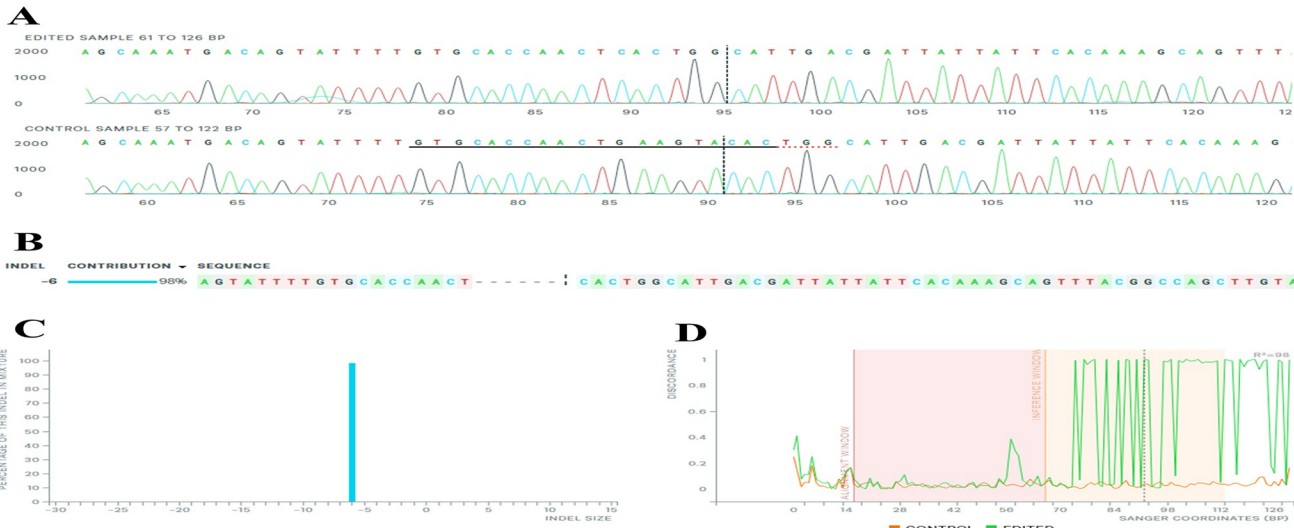

**Fig 6. The comparison between sequencing results of X2-KO and wild-type populations.** A) Sanger sequencing traces of the X2 (edited sample) and wild-type (control sample) regions around the 20-nt target sequence are shown. The area below is marked with black and red dashed lines representing the 20-nt target and the PAM sequence, respectively. The vertical black dotted line represents the cleavage site. The PAM sequence (TGG) has been intact despite deleting nucleotides upstream of the cleavage site. B) The 6-bps deletion is observed in the X2 monoclonal population, causing a PTC in both alleles of the *EpCAM* gene. This mutation signifies that this population encodes a truncated protein and is a bi-allelic knockout (KO) clone, as confirmed by Sanger sequencing of the genomic DNA region around the Z14 monoclonal population's target site (edited sample), comparing it to the reference wild-type sequence (control sample). C) The InDel bar graph depicts the percentage distribution of deletions within the monoclonal Z14 genome population, categorized by InDel size. D) The alignment window refers to the region of the sequence where the alignment is performed between different sequences, and the inference window is the segment within the alignment window where the inference or assessment of discrepancies is made. The discordance plot illustrates the level of alignment per base between the wild type (control) and the monoclonal X2 population (edited) for each nucleotide position. CRISPR editing function causes discrepancies to appear around the cleavage site, and the alignment lines remain far apart after the cleavage site, indicating a high degree of sequence discordance. *Synthego Performance Analysis, ICE Analysis. 2019. v3.0. Synthego; [2023]*.

CRISPR system delivery. Although plasmid vectors may demonstrate lower efficiency and elicit more cell stress responses compared to ribonucleoproteins (RNPs), they are considered the best choice for a wide range of studies due to their affordability, flexibility, and ease of use. In this study, we utilized HCT-116 as a representative adherent cancer cell line. Researchers can also employ other adherent cell lines of different origins, such as those from lung and breast tissues, to generate cell lines. Although concerns about safety and efficacy due to unintended off-target effects persist, we employed cost-effective methods to mitigate such effects and also provided advanced suggestions for researchers where possible. For suspension cells,

**Table 4. Comparison between sequence of amino acids for CD-326 of X6(allele-1) and wild-type populations.**

| Cell population | Sequence of amino acids for CD-326 |
|---|---|
| Wild-type | MAPPQVLAFGLLLAAATATFAAAQEECVCENYKLAVNCFVNNNRQCQCTSVGAQNTVICSK LAAKCLVMKAEMNGSKLGRRAKPEGALQNNDGLYDPDCDESGLFKAKQCNGTSMCWCVN TAGVRRTDKDTEITCSERVRTYWIIIELKHKAREKPYDSKSLRTALQKEITTRYQLDPKFITSIL YENNVITIDLVQNSSQKTQNDVDIADVAYYFEKDVKGESLFHSKKMDLTVNGEQLDLDPGQ TLIYYVDEKAPEFSMQGLKAGVIAVIVVVVIAVVAGIVVLVISRKKRMAKYEKAEIKEMGEM HRELNA* |
| X6(allele-1) | MAPPQVLAFGLLLAAATATFAAAQEECVCENYKLAVNCFVNNNRQ--CTSVGAQNTVICSK LAAKCLVMKAEMNGSKLGRRAKPEGALQNNDGLYDPDCDESGLFKAKQCNGTSMCWCVN TAGVRRTDKDTEITCSERVRTYWIIIELKHKAREKPYDSKSLRTALQKEITTRYQLDPKFITSIL YENNVITIDLVQNSSQKTQNDVDIADVAYYFEKDVKGESLFHSKKMDLTVNGEQLDLDPGQ TLIYYVDEKAPEFSMQGLKAGVIAVIVVVVIAVVAGIVVLVISRKKRMAKYEKAEIKEMGEM HRELNA* |

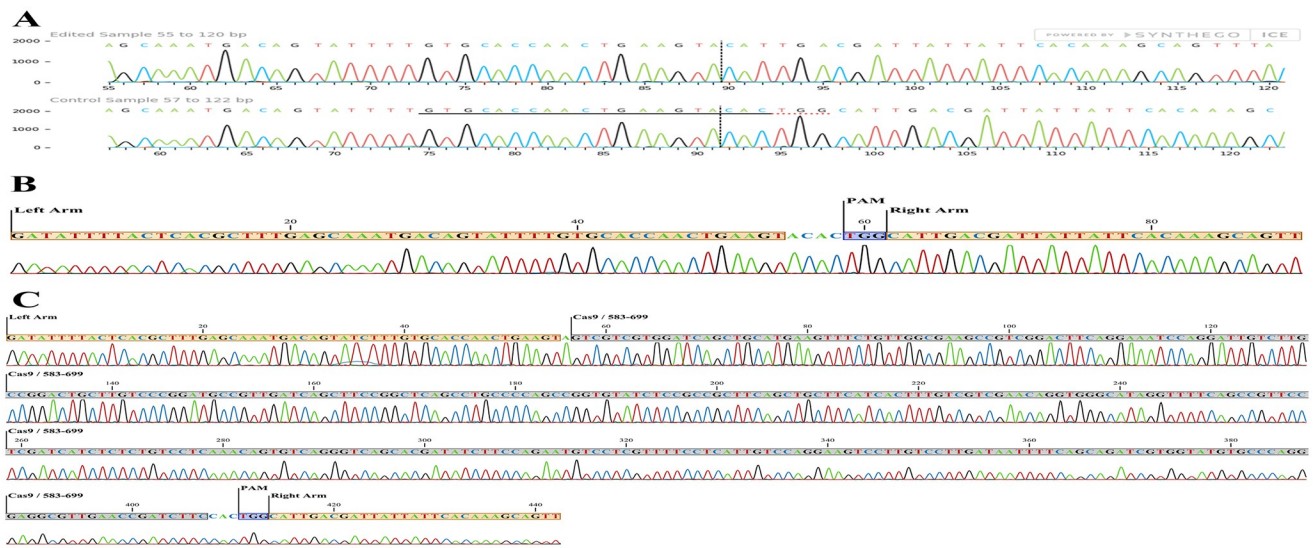

**Fig 7. The comparison between sequencing results of X6 (allele-1), X6 (allele-2) and wild-type populations.** A) Sanger sequencing traces of the X6 (allele-1, edited sample) and wild-type (control sample) regions around the 20-nt target sequence are shown. The area below is marked with black and red dashed lines representing the 20-nt target and the PAM sequence, respectively. The vertical black dotted line represents the cleavage site. The 6-bps deletion is observed in the X2 monoclonal population; the PAM sequence (TGG) and 3-nt upstream of them have been deleted, causing X6 (allele-1) encodes proteins to lack cysteine and glutamine in their structure *Synthego Performance Analysis, ICE Analysis. 2019. v3.0. Synthego; [2023]*. **B, C)** The Sanger sequencing traces of the wild-type (control) and the X6 (allele-2, edited sample) areas around the 20-nt target sequence are shown in B and C, respectively. Compared to the X6(allele-1) sequence, the X6(allele-2) exhibits no deleted nucleotides. Instead, a 351-bp fragment from the Cas9 coding region has been inserted at the cleavage site. The PAM sequence, the sequence of the insert fragment, and the similarities of the sequences are highlighted in blue, gray, and orange, respectively.

researchers can follow the general roadmap described in this protocol, although further optimization and additional steps may be required.

We are aware that financial constraints and limited access to state-of-the-art facilities pose significant challenges for some researchers wishing to conduct studies in personalized medicine. The generation of cell line models is a critical step in evaluating gene functions, yet the high cost of commercial cell line models often acts as a barrier to these studies. We believe that our research paves the way for innovative advancements in gene editing within cancer studies. In conclusion, the primary purpose of this study was to introduce a comprehensive, affordable,

**Table 5. Comparison between sequence of amino acids for CD-326 of X6/ allele-2 and wild-type population.**

| Cell population | Sequence of amino acids for CD-326 |
|---|---|
| Wild-type | MAPPQVLAFGLLLAAATATFAAAQEECVCENYKLAVNCFVNNNRQCQC TSVGAQNTVICSK LAAKCLVMKAEMNGSKLGRRAKPEGALQNNDGLYDPDCDESGLFKAKQCNGTSMCWCVN TAGVRRTDKDTEITCSERVRTYWIIIELKHKAREKPYDSKSLRTALQKEITTRYQLDPKFITSIL YENNVITIDLVQNSSQKTQNDVDIADVAYYFEKDVKGESLFHSKKMDLTVNGEQLDLDPGQ TLIYYVDEKAPEFSMQGLKAGVIAVIVVVVIAVVAGIVVLVISRKKRMAKYEKAEIKEMGEM HRELNA* |
| X6/ allele-2 | MAPPQVLAFGLLLAAATATFAAAQEECVCENYKLAVNCFVNNNRQCQWSSWISCMKFLLA KPSDFRKSRIVLPDCLSRMPLISFRLSLPQPVYLRRFSCFITLSSNRWA*VFSRSSIISLSSNSVRV STISSRMSSFSSSLSRKSLSLIIFSRSWYVPREALNRSSTSVGAQNTVICSK LAAKCLVMKAEMNGSKLGRRAKPEGALQNNDGLYDPDCDESGLFKAKQCNGTSMCWCVN TAGVRRTDKDTEITCSERVRTYWIIIELKHKAREKPYDSKSLRTALQKEITTRYQLDPKFITSIL YENNVITIDLVQNSSQKTQNDVDIADVAYYFEKDVKGESLFHSKKMDLTVNGEQLDLDPGQ TLIYYVDEKAPEFSMQGLKAGVIAVIVVVVIAVVAGIVVLVISRKKRMAKYEKAEIKEMGEM HRELNA* |

**Table 6. Characterization of insert fragment.**

| Name of insert fragment | Sequence of amino acids of insert fragment |
|---|---|
| Partial coding region of Cas9 | EDRFNASLGTYHDLLKIIKDKDFLDNEENEDILEDIVLTLTLFEDREMIEERLKTYAHLF DDKVMKQLKRRRYTGWGRLSRKLINGIRDKQSGKTILDFLKSDGFANRNFMQLIHDD |

and effective method for generating knockout cell lines, leveraging the unparalleled capabilities of the CRISPR/Cas9 system. This approach has immense potential to expedite cancer research by unraveling gene functionalities, elucidating disease mechanisms, and pinpointing potential therapeutic targets. The invaluable insights gleaned from our work promise to steer the development of precision medicine in oncology and promote the development of innovative therapeutic approaches.

## Supporting information

**S1 Fig. Cultivation process upon transformation.** A) Pouring homogenized suspension onto LB-Agar plate: 1) Prepare LB-agar plates containing 100 µg/mL ampicillin. 2) Homogenize bacterial suspensions for both the transformed cells and negative control. 3) Carefully pour each homogenized bacterial suspension into the center of the LB-agar plate, ensuring a gentle pour to prevent splashing or uneven distribution. B) Evenly Distributing the Suspension Using a Sterile Disposable Spreader: 4) Pour onto the agar plate using a sterile disposable spreader following suspension. 5) Start from the center where the suspension was poured; gently move the spreader back and forth to distribute the bacterial suspension evenly at the agar's surface. Maintain a gentle and smooth spreading motion to prevent damage to the agar surface or establish uneven bacterial colonies. Created with BioRender.com.
(TIF)

**S2 Fig. Gel electrophoresis results of the PCR products from seven distinct colonies (C1 to C7).** Each PCR product exhibits a distinct 273-bp band, confirming the accurate cloning of the sgRNA. The term 'NC' denotes the negative control, showing no bands. A ladder reference, ranging from 100 to 1500 bps, was employed, with bands separated by 100-bp intervals. Notably, the highest band appears at 500 bps beyond the 1000-bp band. This outcome strongly supports the successful sgRNA cloning process within the recombinant pX459 plasmids.
(TIF)

**S3 Fig. Results of the MTT assay.** A) 96-Well Plate for the ELISA absorbance measurements. Each column represents distinct concentrations of puromycin antibiotic treatment ranging from 0 to 8 µg/ml. Specific concentrations are labeled at the column base, while 'B' denotes the blank value. A stronger purple coloration indicates a high level of living cells. B) Puromycin kill curve. We measured absorbances at 570 nm and 630 nm for seven replicates of each concentration of puromycin treatment. The results can be found in the S6 and S7 Tables. We imported the final raw data (provided in S8 Table) into GraphPad Prism for analysis and to draw the puromycin kill curve for the HCT-116 cell line, which received a 48-hour puromycin treatment. The graph shows a negative correlation between puromycin concentration and cell viability. The desired concentration is defined as the lowest amount of puromycin at which cell viability reaches zero. Our results showed that 3 µg/ml puromycin is the minimum concentration required to achieve complete death of untransfected cells. This chart also includes standard deviation values; for some concentrations, the standard deviation is less than 1 percent, making it difficult to detect on this scale. For concentrations such as 0 µg/ml (negative control) and 2 µg/ml, which exhibit higher standard deviations, detection is easier. All survival

percentages and corresponding standard deviations for each concentration are listed in the S9 Table.
(TIF)

**S4 Fig. Transfection pattern in a 24-well plate.** Columns 1 and 2 contain three replicates of Complex-1 and Complex-2, respectively. Column 3 represents the negative control, devoid of a recombinant plasmid and containing only lipofectamine. This control group is used to evaluate the probable side effects of lipofectamine during transfection. Column 4 represents the null transfection control, untreated with any complex. This control is used to verify the accuracy of puromycin selection after transfection. The control groups (negative control and null transfection) are placed in separate columns to facilitate easy comparison and analysis of the experimental results. Created with BioRender.com.
(TIF)

**S5 Fig. HCT-116 cell morphologies during transient transfection using lipofectamine.** A) At the initiation of transfection, HCT-116 cells demonstrate a confluence of 50–60%. B) At 48 hours post-transfection, cells treated with Complex-1 exhibit reduced elongated shapes compared to their initial state. C) At 48 hours post-transfection, cells treated with negative-complex display a remarkable increase in confluence, reaching 90–100%. Dense cell junctions are prominently observed. D) Cells in the no-transfection well reach complete confluence 48 hours post-transfection. All observations were obtained using a 20× objective lens. E) After a 48-hour puromycin selection, the complex-1 well displays numerous transfected cells. F) Conversely, cells in the no-transfection well after 48-hour puromycin selection exhibit complete cell death. Visualizations for (E) and (F) were obtained using a 10× objective lens.
(TIF)

**S6 Fig. Sanger sequencing results of wild-type (unedited) and transfected (edited) samples.** A) The sequencing result of the HCT-116 wild-type population illustrates the regular and unique peaks of sequence arrangement. The PAM and the 20-nt target sequences are highlighted in green and yellow, respectively. The cleavage site is located between nucleotides A and C, which is marked in red. B, C) Images B and C indicate the sequencing results of cell populations treated with complex-1 and complex-2, respectively. The NHEJ mechanism induces different repairs at the cleavage sites, resulting in varied sequences between cells. Consequently, multiple nucleotide variations emerge at unique locations, manifesting as irregular short peaks in the sequencing data. In contrast to the wild-type population containing both the PAM and the 20-nt target sequence, which are concurrently present, these sequences are not simultaneously identified within populations treated with complex-1 and complex-2. Overall, this result provides a comprehensive visual of the correct performance of the CRISPR system, elucidating the genetic modifications in complex-1 and complex-2 compared to the wild-type population.
(TIF)

**S7 Fig. Expansion of isolated cells over three weeks.** A, B, C) Z-14 single cell population. Images A, B, and C exhibit the Z-14 single-cell population after 7, 14, and 21 days of isolation, respectively. This single cell was attached to the corner of the well after isolation. D, E, F) X-2 single cell population. Images D, E, and F showcase the X-2 single-cell population after 7, 14, and 21 days of isolation, respectively. Similar to the Z-14 population, these images illustrate the progression of single-cell proliferation over the three weeks. G, H, I) X-7 Cell Population. Images G, H, and I present the X-7 cell population after 7, 14, and 21 days, respectively. Notably, the X-7 well initially received two cells during the isolation process. These images vividly

depict the high proliferation of population due to originating from a multi-cellular state. All images were captured using a 10x objective lens.
(TIF)

**S8 Fig. Gel electrophoresis result of PCR products for six monoclonal cell populations and the wild-type HCT-116 cell line.** Performing gel electrophoresis is necessary to evaluate PCR product quality for Sanger sequencing. All samples, except for X6, exhibited a singular band at around 180 bps. The X6 sample displayed an additional band ranging between 500 and 600 bps. We purified both bands from the gel and executed Sanger sequencing. The insertion of a fragment from the plasmid containing the Cas9 coding region into the cleavage site of one allele was validated by further analysis. Notably, 'NC' denotes the negative control, showing no bands. A 100–1500 bps ladder was utilized, with bands spaced 100 bps apart, while the highest band was 500 bps distant from the 1000 bps band.
(TIF)

**S1 File. Step-by-step protocol, also available on protocols.io.**
(PDF)

**S2 File. EpCAM Sanger sequencing traces for the wild-type HCT-116 cell line.**
(AB1)

**S3 File. EpCAM Sanger sequencing traces for the Z14-KO HCT-116 cell line.**
(AB1)

**S4 File. EpCAM Sanger sequencing traces for the X2-KO HCT-116 cell line.**
(AB1)

**S5 File. EpCAM Sanger sequencing traces for the X6 (allele-1) HCT-116 cell line.**
(AB1)

**S6 File. EpCAM Sanger sequencing traces for the X6 (allele-2) HCT-116 cell line.**
(AB1)

**S7 File. EpCAM Sanger sequencing traces for the transfected (complex-1) sample.**
(AB1)

**S8 File. EpCAM Sanger sequencing traces for the transfected (complex-2) sample.**
(AB1)

**S1 Table. Comprehensive troubleshooting.**
(DOCX)

**S2 Table. Designed gRNAs using bioinformatics tools for the *EPCAM* gene.**
(DOCX)

**S3 Table. Preparation of Puromycin solutions with different concentrations.**
(DOCX)

**S4 Table. The distribution pattern of eleven different puromycin concentrations in the 96-well plate with seven repetitions.**
(DOCX)

**S5 Table. The distribution pattern of culture medium containing 0.5mg/mL MTT and Puromycin in the 96-well plate.**
(DOCX)

**S6 Table. Results of the absorbance measurement at 570 nm.**
(DOCX)

**S7 Table. Results of the absorbance measurement at 630 nm.**
(DOCX)

**S8 Table. Final results of the absorbance measurement (the replaced values are highlighted in grey).**
(DOCX)

**S9 Table. The average survival rate at different concentrations of puromycin (the selected value is highlighted in grey).**
(DOCX)

## Acknowledgments

We greatly appreciate the assistance of Prof. Seyed Davar Siadat, from the Pasteur Institute of Iran.

## Author Contributions

**Conceptualization:** Hossein Fahimi.

**Formal analysis:** Seyed Alireza Mousavi Kahaki.

**Funding acquisition:** Seyed Alireza Mousavi Kahaki.

**Investigation:** Seyed Alireza Mousavi Kahaki.

**Methodology:** Seyed Alireza Mousavi Kahaki, Nayereh Ebrahimzadeh.

**Project administration:** Arfa Moshiri.

**Supervision:** Hossein Fahimi, Arfa Moshiri.

**Writing – original draft:** Seyed Alireza Mousavi Kahaki.

**Writing – review & editing:** Hossein Fahimi.

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
