## [Decision Letter · Decision Letter 0]

10 Jul 2024

PONE-D-24-24563Development of an optimized protocol for generating knockout cancer cell lines using the CRISPR/Cas9 system, with emphasis on transient transfectionPLOS ONE

Dear Dr. Fahimi,

Thank you for submitting your manuscript to PLOS ONE. After careful consideration, we feel that it has merit but does not fully meet PLOS ONE’s publication criteria as it currently stands. Therefore, we invite you to submit a revised version of the manuscript that addresses the points raised during the review process.

We look forward to receiving your revised manuscript.

Kind regards,

Jian Wu, M.D, Ph.D

Academic Editor

PLOS ONE

Journal Requirements:

3. Please ensure that you refer to Figure 2, 3, and 7 in your text as, if accepted, production will need this reference to link the reader to the figure.

4. When you submit your revision, please provide a PDF version of your protocol as generated by protocols.io (the file will have the protocols.io logo in the upper right corner of the first page) as a Supporting Information file. The filename should be S1_file.pdf, and you should enter “S1 File” into the Description field. Any additional protocols should be numbered S2, S3, and so on. Please also follow the instructions for Supporting Information captions [https://journals.plos.org/plosone/s/supporting-information#loc-captions]. The title in the caption should read: “Step-by-step protocol, also available on protocols.io.”

Please assign your protocol a protocols.io DOI, if you have not already done so, and include the following line in the Materials and Methods section of your manuscript: “The protocol described in this peer-reviewed article is published on protocols.io (https://dx.doi.org/10.17504/protocols.io.[...]) and is included for printing purposes as S1 File.” You should also supply the DOI in the Protocols.io DOI field of the submission form when you submit your revision.

If you have not yet uploaded your protocol to protocols.io, you are invited to use the platform’s protocol entry service [https://www.protocols.io/we-enter-protocols] for doing so, at no charge. Through this service, the team at protocols.io will enter your protocol for you and format it in a way that takes advantage of the platform’s features. When submitting your protocol to the protocol entry service please include the customer code PLOS2022 in the Note field and indicate that your protocol is associated with a PLOS ONE Lab Protocol Submission. You should also include the title and manuscript number of your PLOS ONE submission.

Reviewers' comments:

Reviewer's Responses to Questions

**Comments to the Author**

1. Does the manuscript report a protocol which is of utility to the research community and adds value to the published literature?

Reviewer #1: No

Reviewer #2: Yes

Reviewer #3: Yes

2. Has the protocol been described in sufficient detail?

To answer this question, please click the link to protocols.io in the Materials and Methods section of the manuscript (if a link has been provided) or consult the step-by-step protocol in the Supporting Information files.

The step-by-step protocol should contain sufficient detail for another researcher to be able to reproduce all experiments and analyses.

Reviewer #1: Partly

Reviewer #2: Yes

Reviewer #3: Yes

3. Does the protocol describe a validated method?

Reviewer #1: No

Reviewer #2: Yes

Reviewer #3: Yes

4. If the manuscript contains new data, have the authors made this data fully available?

Reviewer #1: No

Reviewer #2: N/A

Reviewer #3: Yes

**5. Is the article presented in an intelligible fashion and written in standard English?**

Reviewer #1: Yes

Reviewer #2: Yes

Reviewer #3: Yes

6. Review Comments to the Author

Reviewer #1: The overall description of the protocol is satisfactory; however, it appears to be a classical CRISPR-based transient knockout protocol without any evident novelty.

There are several issues that need to be addressed:

Figure Resolution: The resolution of the main figures, particularly Figures 4 to 6, is inadequate. It is difficult to discern the bases in each figure due to the poor quality.

Figure Consistency: The individual photos in Figures 5 and S7, as well as Figure 7, are not in the same frame shape and lack a consistent background. This inconsistency should be corrected for a more professional presentation.

Figure Legend: The legend for Figure 7 is missing and needs to be provided to ensure clarity and proper understanding of the figure.

Validation of Knockout: For the validation part of the model construction, it is crucial to confirm the knockout effect of the target protein EpCAM. A Western blot is necessary to verify that EpCAM has been knocked out in the clones that are claimed to be successfully constructed.

Reviewer #2: The title of the manuscript is clear and accurately reflects the content of the study. The abstract provides a concise overview, highlighting the significance, methodology, and key findings. However, it would benefit from a brief mention of the specific challenges addressed by the new protocol compared to existing methods, such as reducing off-target effects and improving transfection efficiency in various cancer cell lines.

The introduction could be more concise in sections discussing the historical background of CRISPR to improve readability. For example, the discussion on the differences between CRISPR and other genome editing tools like ZFNs and TALENs could be condensed. More emphasis should be placed on the specific gaps in current methodologies that this new protocol aims to address, such as the limitations of viral delivery methods and the challenges of achieving high-efficiency transient transfection in cancer cell lines.

The methods section is detailed and provides step-by-step instructions, which are crucial for reproducibility. The inclusion of troubleshooting tips and the use of robust bioinformatics tools are commendable. However, the manuscript would benefit from a more explicit comparison of the new protocol's effectiveness against existing protocols. Maybe emphasizethe key steps (the comparasions with previous steps) in the protocol in the a flowchart (fig.7) for quick reference would improve accessibility, especially for new researchers.

The results section presents clear evidence of the successful creation of knockout cell lines using the new protocol, and the inclusion of sequencing data to confirm biallelic knockout is a strong point. However, the results could be strengthened by including comparative data showing the performance of the new protocol against established methods, such as viral transfection techniques or other non-viral methods. Providing additional quantitative data on transfection efficiency, cell viability, and off-target effects would enhance the robustness of the findings. For example, including data on the percentage of cells successfully transfected and the frequency of off-target mutations would provide a more comprehensive evaluation of the protocol's effectiveness.

The discussion provides a good interpretation of the results and their implications for cancer research, highlighting the potential applications and benefits of the new protocol in various research contexts. However, it could benefit from a deeper analysis of the limitations of the new protocol and potential areas for further optimization. For instance, discussing the potential integration of plasmid fragments into the host genome and the associated risks in more detail would provide a more balanced view. Additionally, a more detailed comparison with other non-viral delivery methods, such as electroporation or microinjection, could provide clearer insights into the advantages and drawbacks of the lipid nanoparticle-based delivery strategy used in this study.Additionally, discussing the applicability of the protocol to other types of cancer cell lines beyond colorectal cancer, such as breast or lung cancer lines, could broaden its impact and relevance.

The figures and tables included in the manuscript are well-organized and support the text effectively. However, some figures could benefit from more detailed captions explaining the experimental setup and results. For example, the figure 7 illustrating the steps of the protocol should include more descriptive captions that detail the significance of each step. Including error bars and statistical analysis in the figures would provide a better understanding of the data's reliability, enhancing the scientific rigor of the presented results.

Overall, the manuscript is well-written and makes a valuable contribution to the field of genome editing. Attention to minor grammatical errors and typos will enhance the overall readability. It could be expanded to include a brief discussion on the potential impact of this protocol on clinical applications and personalized medicine. Addressing these specific areas for improvement will strengthen the manuscript and enhance its contribution to the field.

Reviewer #3: In this protocol, Seyed and other authors described a detailed protocol to generate knockout cell lines using the CRISPR/Cas9 technology by transient transfection. The authors used the EpCAM gene as an example to provide a step-by-step protocol for deleting a gene in a cancer cell line. The protocol is well-written and easy to understand for persons without much experience in the field. Overall, I would like to thank the authors for their efforts in providing such a detailed protocol. Meanwhile, my general comment is that this protocol contains too many details that might be distractive. Below are my specific comments for the authors.

1. The abstract can be shorter to emphasize the contents of the current protocol since CRISPR is a well-known technology already.

2. Lines 51-60 talked about the off-target effect of CRISPR. Does off-target effect is the main focus of this protocol? Please provide rationale and references regarding to why this protocol can minimize the off-target effects.

3. Line 472-500. It seems that all the lab consumables were listed in the equipment session. My general impression is that materials like pipette tips (especially the colors) don’t have to be described.

4. Since the authors mentioned that this protocol can help researchers with limited funding opportunities. They may want to consider pointing out which reagents are critical to success and which are dispensable or interchangeable between vendors.

5. Some preparation methods for commonly used reagents may not have to be described. For example, line 504, ampicillin solution.

6. To increase the readability of the protocol: Standard experimental conditions don't have to be described, for example, 37 °C with 5% CO2 for cell culture. If the reagents and volumes have been displayed in the table, please consider to reduce the verbal description (For example in STEP 48).

7. PLOS authors have the option to publish the peer review history of their article (what does this mean?). If published, this will include your full peer review and any attached files.

Reviewer #1: No

Reviewer #2: No

Reviewer #3: No

---

## [Author Response · Author response to Decision Letter 0]

16 Aug 2024

Dear Editor,

We have carefully reviewed your comments and addressed the concerns regarding the requested items. One of the respected reviewers (Reviewer 1) mentioned in Comment 1 that some images appeared to be of low quality. However, after performing a double-check, we confirm that all images were uploaded following PLOS ONE requirements, using the PACE tool to ensure the highest possible quality. It seems that the submission system’s compression process may have affected the image quality, which is beyond our control. Additionally, we have thoroughly addressed the reviewers’ comments in the response provided below. The response to the reviewers’ comments has also been uploaded in the attached files section.

Reviewer #1:

Reviewer’s Comment 1: The resolution of the main figures, particularly Figures 4 to 6, is inadequate. It is difficult to discern the bases in each figure due to the poor quality.

Auothers’ Response: We meticulously reviewed the reported issues and can confirm that all figures, both main and supporting, were uploaded at the highest permissible resolution according to the journal’s requirements. Additionally, we used the Preflight Analysis and Conversion Engine (PACE) digital diagnostic tool to ensure that the figures meet PLOS requirements. After thorough checks, we can confidently declare that both the original and reformatted sources present all pictures in good quality with no undetectable issues. It appears that a compression process within the submission system may have affected the image quality, which is beyond our control. We assure you that we will promptly report this issue, along with this response, to the journal, and collaborate with them to resolve the reported issue.

Reviewer’s Comment 2: The individual photos in Figures 5 and S7, as well as Figure 7, are not in the same frame shape and lack a consistent background. This inconsistency should be corrected for a more professional presentation.

Auothers’ Response: We did our best to achieve a professional presentation and strived to present our results at the highest possible quality while minimizing the mentioned issues. It is important to note that we conducted this study with minimal funding support and limited access to professional equipment. The background inconsistencies in the cell images are due to our equipment’s limitations. Specifically, we had to use an external camera through the eyepiece lens because our microscope lacked an integrated camera for direct image capture. Despite our efforts to obtain high-quality images, our conventional camera lacked advanced features such as smart focus settings and optical stabilization, and varying transparency at different magnifications further intensified the observed inconsistencies..

Reviewer’s Comment 3: Figure Legend: The legend for Figure 7 is missing and needs to be provided to ensure clarity and proper understanding of the figure.

Auothers’ Response: After reviewing the addressed issue, we confirm that the legend for Figure 7 was located between the discussion and acknowledgments sections. Additionally, we have revised the legend to include more details in the description to improve clarity and ensure proper understanding of the figure.

Reviewer’s Comment 4: Validation of Knockout: For the validation part of the model construction, it is crucial to confirm the knockout effect of the target protein EpCAM. A Western blot is necessary to verify that EpCAM has been knocked out in the clones that are claimed to be successfully constructed.

Auothers’ Response: We acknowledge that Western blotting is the gold standard for validating changes in protein levels. Unfortunately, the antibodies required for Western blotting had to be ordered from abroad, as they were not available in Iran. The strict shipping policies imposed by the Iranian government during the COVID-19 pandemic resulted in extraordinary shipping fees for biological materials, with no guarantee of proper shipment conditions, particularly for those requiring cold storage. During this period, numerous studies in Iran faced significant challenges, as they either failed to receive the ordered materials or received biological materials that had been spoiled due to inappropriate shipping conditions. Despite our best efforts, logistical and financial limitations made performing Western blotting impossible. Consequently, we used the Sanger sequencing validation method for our knockout clones. Certain applications of CRISPR, such as knockdown and overexpression, do not alter the DNA sequence and require validation at the RNA or protein levels. We validated our protocol’s performance and CRISPR’s function by sequencing the target region. Additionally, after consulting with various biological companies, including Horizon, we learned that many commercially available knockout cell lines are often validated using Sanger sequencing for quality control. We have provided a certificate of analysis for a knockout cell line that is sold by a biological company in this link (Product link). 

Reviewer #2: 

Reviewer’s Comment 1:The title of the manuscript is clear and accurately reflects the content of the study. The abstract provides a concise overview, highlighting the significance, methodology, and key findings. However, it would benefit from a brief mention of the specific challenges addressed by the new protocol compared to existing methods, such as reducing off-target effects and improving transfection efficiency in various cancer cell lines.

Auothers’ Response: We have revised the abstract and made significant changes based on suggestions to make it more concise and emphasize the study’s focus. We removed background information on CRISPR and unnecessary data to improve readability. As a result, the new abstract clearly and concisely focuses on methodology and findings. We have mentioned that we developed an approach employing a unique strategy by adhering to strict criteria for designing highly effective gRNAs, coupled with transient transfection, to reduce off-target effects. Comprehensive details, along with step-by-step instructions, are provided in the related sections. Our goal in this study was to introduce an optimized approach to minimize off-target effects based on previous prestigious investigations rather than measure off-target effects for different approaches. Given that the suggestions for enhancing transfection efficiency are detailed, we have included them in the experimental design section of the manuscript. Additionally, we mention that the choice of transfection method is entirely dependent on cell type characteristics, and further optimization or alternative approaches may be required for certain difficult-to-transfect cells. According to our research, Lipofectamine 3000 demonstrates high transfection efficiency for the vast majority of cell lines, particularly for difficult-to-transfect cells, which is why we selected this reagent for our studies.

Reviewer’s Comment 2:The introduction could be more concise in sections discussing the historical background of CRISPR to improve readability. For example, the discussion on the differences between CRISPR and other genome editing tools like ZFNs and TALENs could be condensed. More emphasis should be placed on the specific gaps in current methodologies that this new protocol aims to address, such as the limitations of viral delivery methods and the challenges of achieving high-efficiency transient transfection in cancer cell lines.

Auothers’ Response: Following the suggestion, we have completely rewritten the "Introduction" to remove unnecessary information, such as CRISPR's historical data and comparisons with other genome editing tools like ZFNs and TALENs, to improve readability by avoiding irrelevant descriptions. We also added a section with references that briefly discusses the main problems with stable transfection, such as the increased risk of off-target effects and insertional mutagenesis associated with lentiviral delivery methods. The “Experimental design” section provides a detailed description of the most challenging factors that significantly affect transient transfection efficiency. Because this protocol contains many details, we organized the manuscript by categorizing the content into specific topics to prevent confusion and enhance understanding. These topics include “Development of the protocol,” “Comparison with other CRISPR genome editing methods,” “Limitations of using this protocol,” and “CRISPR/Cas9 delivery methods.” In the “Experimental design” section, we have addressed solutions to tackle common challenges. These solutions include using stringent criteria to minimize off-target effects, employing important factors to increase transient transfection efficiency, and providing suggestions for improving single-cell isolation yields.

Reviewer’s Comment 3:The methods section is detailed and provides step-by-step instructions, which are crucial for reproducibility. The inclusion of troubleshooting tips and the use of robust bioinformatics tools are commendable. However, the manuscript would benefit from a more explicit comparison of the new protocol's effectiveness against existing protocols. Maybe emphasizethe key steps (the comparasions with previous steps) in the protocol in the a flowchart (fig.7) for quick reference would improve accessibility, especially for new researchers.

Auothers’ Response: We have added more details to the “Comparison with other CRISPR genome editing methods” section, demonstrating the advantages of using this protocol compared to other approaches, particularly those employing stable transfection. Additionally, the “Development of the protocol” section concisely describes the main advantages, noting that while the previous protocol outlined general guidelines and suggested various methods for conducting CRISPR projects, it lacked essential guidance, especially for researchers undertaking CRISPR projects for the first time. To further enhance accessibility, we have revised the legend of the flowchart and made significant changes that provide detailed explanations for each step of the flowchart, offering a comprehensive overview at a glance. Furthermore, the professional troubleshooting table addresses almost all potential issues, thus avoiding wasted time in finding solutions. This is especially important for new researchers, enabling them to save time and focus more on their goals.

Reviewer’s Comment 4:The results section presents clear evidence of the successful creation of knockout cell lines using the new protocol, and the inclusion of sequencing data to confirm biallelic knockout is a strong point. However, the results could be strengthened by including comparative data showing the performance of the new protocol against established methods, such as viral transfection techniques or other non-viral methods. Providing additional quantitative data on transfection efficiency, cell viability, and off-target effects would enhance the robustness of the findings. For example, including data on the percentage of cells successfully transfected and the frequency of off-target mutations would provide a more comprehensive evaluation of the protocol's effectiveness.

Auothers’ Response: We acknowledge that the suggestions described here would significantly enhance the robustness of our findings. To present comprehensive comparative data, it was essential to run all possible modes with specified variables under similar conditions. For example, to compare the off-target effects between stable and transient transfection, both experiments needed to be conducted under identical conditions to ensure reliable results. However, due to severe funding limitations and lack of access to modern facilities, we were unable to expand our studies to perform such comparative analyses. Instead, we designed this study based on findings from validated previous investigations. Our aim was to create a straightforward protocol that researchers in similar circumstances can use to generate different knockout cell lines for evaluating gene function in cancer investigations, despite these limitations.Here are some examples that could address the concerns raised:

1. Comparison with viral vectors and other non-viral methods: While it is evident that some viral vectors exhibit high transfection efficiency for both in vivo and in vitro studies, they also have significant drawbacks. Unpredictable integration into the host genome can disrupt tumor suppressor genes or promote proto-oncogene expression due to their strong promoters. This, along with high levels of induced stress and capacity limitations, poses serious challenges for optimizing them across a wide spectrum of clinical studies. Other non-viral methods, such as electroporation and microinjection, though effective, face limitations for in vivo studies, leading to varied outcomes under different conditions and presenting significant challenges for clinical applications. Although comparing different methods provides valuable insights, many papers already highlight these differences. Our main focus was on designing instructions with high validity and minimal requirements, considering our funding limitations and the inaccessibility of modern facilities.

2. Transfection Efficiency Measurement: The inaccessibility of relevant equipment was the first barrier to measuring transfection efficiency, and the type of vector was another important factor. There are two similar vectors with different selectable markers: pX458 with GFP and pX459 with a puromycin selectable marker. For this study, we used pX459 and determined the optimal puromycin concentration to eliminate untransfected cells. If the MTT assay, which finds the best puromycin concentration, is done wrong, either untransfected cells could live or transfected cells could die from a too-high puromycin dose, making it impossible to calculate the transfection efficiency for comparison purposes. Several factors, including the passage number of cell lines, penicillin-streptomycin antibiotics, lipopolysaccharide (LPS) contamination in the extracted recombinant plasmid, cell confluence percentage, and serum content in the cell culture medium, among others, may have negligible or significant effects individually or collaboratively. These factors require further study to fully understand their impact using a vector with a fluorescent tag like pX458. We suggested the optimal conditions based on previous studies in the Experimental Design section. Additionally, the transfection efficiency of Lipofectamine 3000 for the most commonly used cell line models is indicated on the manufacturer’s website.

3. Off-Target Effects: Reducing off-target effects as much as possible was one of the major goals when designing this study, as ensuring that our manipulations do not affect unintended regions is crucial for enhancing the validity of our results, especially for clinical applications. We utilized the most robust bioinformatic tools available in this area. Although there is a comprehensive library containing pre-designed gRNA, we did not suggest using pre-designed gRNA or a single bioinformatic tool for gRNA design. Each bioinformatics tool uses a different algorithm for gRNA design and may have varying criteria for predicting off-target effects, sometimes overlooking important factors. Therefore, we introduced the current best tools, used all of them, and selected a gRNA recommended by all tools with the highest rank for on-target efficiency and minimal off-target potential. Across all tools, our selected gRNA did not indicate dangerous predicted off-target effects on functional genes.

Reviewer’s Comment 5:The discussion provides a good interpretation of the results and their implications for cancer research, highlighting the potential applications and benefits of the new protocol in various research contexts. However, it could benefit from a deeper analysis of the limitations of the new protocol and potential areas for further optimization. For instance, discussing the potential integration of plasmid fragments into the host genome and the associated risks in more detail would provide a more balanced view. Additionally, a more detailed comparison with other non-viral delivery 

---

## [Decision Letter · Decision Letter 1]

30 Aug 2024

Development of an optimized protocol for generating knockout cancer cell lines using the CRISPR/Cas9 system, with emphasis on transient transfection

PONE-D-24-24563R1

Dear Dr. Fahimi,

We’re pleased to inform you that your manuscript has been judged scientifically suitable for publication and will be formally accepted for publication once it meets all outstanding technical requirements.

Kind regards,

Jian Wu, M.D, Ph.D

Academic Editor

PLOS ONE

Additional Editor Comments (optional):

Reviewers' comments:

Reviewer's Responses to Questions

**Comments to the Author**

1. Does the manuscript report a protocol which is of utility to the research community and adds value to the published literature?

Reviewer #1: Yes

Reviewer #2: Yes

Reviewer #3: Yes

2. Has the protocol been described in sufficient detail?

To answer this question, please click the link to protocols.io in the Materials and Methods section of the manuscript (if a link has been provided) or consult the step-by-step protocol in the Supporting Information files.

The step-by-step protocol should contain sufficient detail for another researcher to be able to reproduce all experiments and analyses.

Reviewer #1: Yes

Reviewer #2: Yes

Reviewer #3: Yes

3. Does the protocol describe a validated method?

Reviewer #1: Yes

Reviewer #2: Yes

Reviewer #3: Yes

4. If the manuscript contains new data, have the authors made this data fully available?

Reviewer #1: Yes

Reviewer #2: Yes

Reviewer #3: Yes

**5. Is the article presented in an intelligible fashion and written in standard English?**

Reviewer #1: Yes

Reviewer #2: Yes

Reviewer #3: Yes

6. Review Comments to the Author

Reviewer #1: There are lots of difficulties for the authors to do research in Iran.But personally speaking, it shouldn't be the acceptable reason.

Reviewer #2: I have reviewed the revised manuscript. The revisions effectively addressed all the concerns raised in the initial review.

Reviewer #3: I have no further questions. My previous questions have been sufficiently addressed in the revised manuscript.

7. PLOS authors have the option to publish the peer review history of their article (what does this mean?). If published, this will include your full peer review and any attached files.

Reviewer #1: No

Reviewer #2: No

Reviewer #3: No

---

## [Editor Report · Acceptance letter]

5 Nov 2024

PONE-D-24-24563R1 

PLOS ONE

Dear Dr. Fahimi, 

I'm pleased to inform you that your manuscript has been deemed suitable for publication in PLOS ONE. Congratulations! Your manuscript is now being handed over to our production team.

Kind regards, 

on behalf of

Dr. Jian Wu 

Academic Editor

PLOS ONE